# Numerical Investigation on the Effect of Section Width on the Performance of Air Ejector with Rectangular Section

**DOI:** 10.3390/e25010179

**Published:** 2023-01-16

**Authors:** Ying Zhang, Jingming Dong, Shuaiyu Song, Xinxiang Pan, Nan He, Manfei Lu

**Affiliations:** 1Marine Engineering College, Dalian Maritime University, Dalian 116026, China; 2College of Ocean Engineering, Guangdong Ocean University, Zhanjiang 524088, China

**Keywords:** air ejector, rectangular section, turbulent kinetic energy, shock train, vorticity

## Abstract

Due to its simple structure and lack of moving parts, the supersonic air ejector has been widely applied in the fields of machinery, aerospace, and energy-saving. The performance of the ejector is influenced by the flow channel structure and the velocity of the jet, thus the confined jet is an important limiting factor for the performance of the supersonic air ejector. In order to investigate the effect of the confined jet on the performance of the ejector, an air ejector with a rectangular section was designed. The effects of the section width (*W_c_*) on the entrainment ratio, velocity distribution, turbulent kinetic energy distribution, Mach number distribution, and vorticity distribution of the rectangular section air ejector were studied numerically. The numerical results indicated that the entrainment ratio of the rectangular section air ejector increased from 0.34 to 0.65 and the increment of the *ER* was 91.2% when the section width increased from 1 mm to 10 mm. As *W_c_* increased, the region of the turbulent kinetic energy gradually expanded. The energy exchange between the primary fluid and the secondary fluid was mainly in the form of turbulent diffusion in the mixing chamber. In addition to *W_c_* limiting the fluid flow in the rectangular section air ejector, the structure size of the rectangular section air ejector in the XOY plane also had a limiting effect on the internal fluid flow. In the rectangular section air ejector, the streamwise vortices played an important role in the mixing process. The increase of *W_c_* would increase the distribution of the streamwise vortices in the constant-area section. Meanwhile, the distribution of the spanwise vortices would gradually decrease.

## 1. Introduction

An air ejector is a type of pneumatic device, which uses a vacuum created by the primary fluid to entrain the secondary fluid [1,2,3]. A low-speed secondary fluid interacts with the high-speed primary fluid in the mixing chamber to exchange mass, momentum, and energy in the restricted flow channel. Next, the mixing process of the two fluids generates complex gas dynamics phenomena such as shock waves and shear layer inside the air ejector [4,5,6]. The air ejector has a simple structure, no moving parts, and can work without consuming mechanical or electrical energy. The air ejector is an attractive and environmentally friendly device [7], which has been widely used in vacuum systems [8], wind tunnels [9], propulsion devices [10], high-altitude test facilities [11], gas-powered lasers [12], fuel cells [13], and spacesuit portable life support systems [14]. The air ejector has the simple structure, but its operating parameters [15] and structural parameters [16] have significant impacts on the performance of the air ejector. The selection of these parameters is the key to the design of the air ejector, which can directly affect the performance of the air ejector [17]. Mani et al. [18] found that the primary fluid pressure had a great effect on the entrainment ratio of the rectangular ejector through an investigation on the visualization of the shock wave inside the air ejector. Yang et al. [19] had conducted a comparative study on ejectors with circular, rectangular, elliptical, square, and cross-shaped nozzles. According to the results of the comparative study, vortexes with different degrees of reverse rotation appear at the tip of the nozzle outlet. The generation of the vortex will affect the mixing of two fluids in the restricted flow channel and increase the mechanical loss caused by the collision with the wall. Therefore, the vortex has an important influence on the characteristics of the internal flow inside the ejector and the performance of the ejector.

Additionally, the cross-section shape of the ejector and the limitation degree of the restricted flow channel have an important influence on the characteristics of the internal flow inside the ejector and the performance of the ejector. In order to further study the mixing process in the restricted flow channel of the ejector, Bouheraoua et al. [20] conducted a three-dimensional numerical simulation of the rectangular section air ejector. The results showed that in the initial stage of mixing the two fluids were laminar flow, while in the constant-area section they were transit to turbulent. During this process, the vortexes were formed and finally decomposed at the end of the constant-area section. Compared with the free jet, the restricted flow channel could promote the evolution of the confined jet flow field. Under the constraint of the restricted flow channel, the centerline velocity of the confined jet continuously decreased. Simultaneously, its entrainment performance fluctuated [21]. It should also be noted that the restricted flow channel had an important effect on the expansion process of the high-speed jet. The vortex generated at the nozzle corner could enhance the radial expansion of the high-speed jet [22]. The recirculation zone was affected by the structure of the restricted flow channel and the velocity of the jet. Compared with the velocity of the high-speed jet, the structure of the restricted flow channel played a dominant role in the generation of the recirculation zone [23], thus determining the central velocity and expansion of the high-speed jet.

In the restricted flow channel, the mixing process of the primary fluid and secondary fluid was complex [24]. Therefore, the confined jet was an important limiting factor affecting the performance of the ejector [25]. Compared with low-speed gas, supersonic gas had a better mixing effect; however, its potential gas dynamics are still being explored [26]. The double-choking phenomenon was one of the obstacles for studying the performance of the ejector. At present, the correctness of the Fabri-choking model has been confirmed by experimental and numerical investigation [27]. The internal flow of the ejector was often described by measurement of the wall pressure and the visualization experiment. The static pressure measurements of the wall could provide insight into the mixing process. The schlieren visualization experiment could capture the mixing characteristics of the confined jets [28]. However, current studies mainly focus on exploring the simulation method and experimental method of the internal flow inside the rectangular section air ejector, but neglects the changes in the flow characteristics of the air ejector.

To further study the flow state of the high-speed jet in the rectangular restricted flow channel, the three-dimensional numerical simulation of the rectangular section air ejector is carried out. In this paper the effects of section width (*W_c_*) on entrainment ratio, velocity distribution, turbulent kinetic energy distribution, Mach number distribution, and vorticity distribution were studied.

## 2. Methods

### 2.1. Geometric Modeling

The methods of ejector design include constant area mixing [29], constant pressure mixing [30], constant rate of momentum change (CRMC) [31], and constant rate of kinetic energy change (CRKEC) [32]. The one-dimensional constant pressure mixing assumes that the constant pressure mixing occurs in the constant-area section of the ejector. It can obtain a more suitable air ejector performance. In order to obtain the structure parameters more suitable for the rectangular section air ejector, a one-dimensional constant pressure mixing theory was selected in this paper. According to the one-dimensional constant pressure mixing theory, the height of nozzle throat, height of nozzle inlet, height of nozzle outlet, and height of constant-area section for the rectangular section air ejector were designed. At the same time, the length dimension of the ejector could be obtained according to the empirical formula. The three-dimensional structure diagram of the rectangular section air ejector and the structure diagram of the rectangular section air ejector on the XOY plane are shown in Figure 1 and Figure 2, respectively. As shown in Figure 1, the rectangular section air ejector had two secondary fluid inlets. The mass flow rate of the secondary fluid was the sum of the mass flow rate of the upper secondary fluid inlet and the lower fluid inlet. Therefore, the entrainment ratio (*ER*) of the rectangular section air ejector was defined as follows:(1)$$ER=\frac{m_{e1}+m_{e2}}{m_{g}}$$
where *m_g_* was the mass flow rate of the primary fluid, *m*_*e*1_ was the mass flow rate of the upper secondary fluid inlet, and *m*_*e*2_ was the mass flow rate of the lower secondary fluid inlet.

As shown in Figure 2, the nozzle outlet (*X_a_*), the inlet of the constant-area section (*X_c_*_0_), the 1/4 of the constant-area section (*X*_*c*1/4_), the 1/2 of the constant-area section (*X*_*c*1/2_), the 3/4 of the constant-area section (*X*_*c*3/4_), and the outlet of the constant-area section (*X*_*c*1_) were selected as the analytical cross-section. The structure parameters of the rectangular section air ejector are shown in Table 1. When the *W_c_* of the rectangular section air ejector was changed, the restricted degree of high-speed jet in the rectangular confined flow channel changed, thus affecting the *ER* of the rectangular section air ejector and the flow characteristics inside the ejector. In this paper, the *ER* of the rectangular section air ejector and the flow characteristics inside the ejector were investigated when the *W_c_* of the rectangular section air ejector varied from 1 mm to 10 mm.

### 2.2. CFD Modeling

In order to simplify the simulation, several simplifications were made as follows:(1)The air in the rectangular section air ejector was the ideal compressible gas;(2)The wall was a non-slip adiabatic wall;(3)Ignore the temperature change caused by the supersonic flow of gas in the whole process;(4)The mixing process was the constant pressure mixing;(5)Ignore the initial velocity of the primary fluid inlet and the secondary fluid inlet.

The governing equation in the solving process was as follows:

Continuity equation:(2)$$\frac{\partial \rho }{\partial t}+\frac{\partial }{\partial x_{i}}\left(\rho u_{i}\right)=0$$

Momentum equation:(3)$$\frac{\partial }{\partial t}\left(\rho u_{i}\right)+\frac{\partial }{\partial x_{i}}\left(\rho u_{i}u_{j}\right)=-\frac{\partial P}{\partial x_{i}}+\frac{\partial \tau_{ij}}{\partial x_{j}}$$

Energy equation:(4)$$\frac{\partial }{\partial t}\left(\rho E\right)+\frac{\partial }{\partial x_{i}}\left[u_{i}\left(\rho E+P\right)\right]=\vec{\nabla }⟮\alpha_{eff}\frac{\partial T}{\partial x_{i}}⟯+\vec{\nabla }\left[u_{j}{\left(\tau_{ij}\right)}_{eff}\right]$$
with
(5)$$\tau_{ij}=\mu_{eff}⟮\frac{\partial u_{i}}{\partial x_{j}}+\frac{\partial u_{j}}{\partial x_{i}}⟯-\frac{2}{3}\mu_{eff}\frac{\partial u_{k}}{\partial x_{k}}\delta_{ij}$$
(6)$$\rho =\frac{P}{RT}$$
where *ρ* was density, *u* was velocity, *τ* was viscous stress, *μ* was dynamic viscosity, *E* was total energy, *P* was static pressure, and *δ* was the Kronecker delta.

It had been verified by experiments that the k-ω SST model in the numerical simulation was more suitable to describe the performance of the ejector and the flow characteristics in the ejector when the primary fluid was air [33]. Therefore, the k-ω SST model was selected as the turbulence model in this paper.

k-ω SST turbulence model:(7)$$\rho \bar{u_{i}^{\prime }u_{j}^{\prime }}=\mu_{t}\left[\frac{\partial u_{i}}{\partial x_{j}}+\frac{\partial u_{j}}{\partial x_{i}}\right]-\frac{2}{3}\left[\rho k+\mu_{t}\frac{\partial u_{k}}{\partial x_{k}}\right]\delta_{ij}$$
(8)$$\frac{\partial }{\partial t}(\rho k)=\frac{\partial }{\partial x_{j}}\left[\Gamma_{k}\frac{\partial k}{\partial x_{j}}\right]+G_{k}-Y_{k}$$
(9)$$\frac{\partial }{\partial t}(\rho \omega )=\frac{\partial }{\partial x_{j}}\left[\Gamma_{\omega }\frac{\partial \omega }{\partial x_{j}}\right]+G_{\omega }-Y_{\omega }+D_{\omega }$$
where *k* was the turbulent kinetic energy, *ω* was the specific turbulence dissipation rate, Γ was the diffusion rate, *G* was the turbulent kinetic energy generated by the laminar flow velocity gradient, *Y* was the turbulent kinetic energy generated by diffusion, and *D* was the orthogonal divergence term.

Fluent 19.2 was used to solve the rectangular section air ejector model. The primary fluid inlet, the upper secondary fluid inlet, and the lower secondary fluid inlet were all set as “pressure inlet”, while the outlet of the diffuser chamber was set as “pressure outlet”. The other walls of the ejector were set as non-slip adiabatic walls. All temperatures were set at 293 K. The coupled method was selected for the iterative solution. The second-order was adopted for the discretization scheme for pressure. The second-order upwind discretization was applied for the all solution. All residual values of the calculated results were less than 10^−6^.

### 2.3. Validation of Grid Independence

In this paper, ICEM CFD was used to obtain the high-quality three-dimensional grid of the ejector. Since the structure of the rectangular section air ejector was symmetric along the XOZ profile, local axisymmetric meshes were selected to save calculation time. The structural mesh was adopted for meshing. In addition, the local mesh refinement approach was adopted in the region near the nozzle to improve the accuracy of the solution, as shown in Figure 3.

In order to ensure the accuracy of the solution and reduce the amount of computation, the grid independence of the rectangular section air ejector was verified. The number of grids increased from 148,864 to 842,258. The solutions were calculated for five different numbers of grids. Data detection points were set at the axis of the rectangular section air ejector for the validation of grid independence. The nozzle outlet (Point A) and the inlet of the constant-area section (Point B) were selected as the detection point. The velocity and pressure values of the two detection points were obtained from the calculation results. Further, the y+ of the first layer grid were kept at approximately 3 by encrypting the mesh of the wall and the nozzle, which satisfied the conditions for the k-ω SST model [34]. The grid independence validation results of Points A and B are shown in Table 2. In this paper, the rectangular section air ejectors under different *W_c_* were calculated with no less than 279,910 grids, which could reduce the calculation time, while ensuring the accuracy of the calculation.

## 3. Results and Discussion

In the rectangular section air ejector, the mixed fluid will collide with the surrounding walls and cause a large amount of energy loss due to the limitation of *W_c_*. In addition, *W_c_* also affects the expansion state of the primary fluid. When *W_c_* changes, the *ER* and the flow characteristic of the rectangular section air ejector change accordingly. This paper focuses on the effect of *W_c_* on the performance and internal flow of the rectangular section air ejector when the primary fluid pressure (*P_p_*) is 400 kPa, the secondary fluid pressure (*P_s_*) is 100 kPa, and the XOY plane size is fixed.

When *W_c_* increases from 1 mm to 10 mm, the variation of the *ER* of the rectangular section air ejector is shown in Figure 4. As shown in Figure 4, the *ER* of the rectangular section air ejector increases rapidly at first and then fluctuates slightly with the increase of *W_c_*. When *W_c_* is 2 mm, the increment of the *ER* of the rectangular section air ejector starts to decrease. When *W_c_* is 5 mm, the incremental decrease of *ER* of the ejector with rectangular section is negligible, and the increment is only 0.014. When *W_c_* is 9 mm, the *ER* of the rectangular section air ejector reaches a maximum. When *W_c_* increases from 1 mm to 10 mm, the *ER* increases from 0.34 to 0.65, and the increment of the *ER* is 91.2%. This indicates that *W_c_* has a great effect on the performance of the rectangular section air ejector. Therefore, in this paper the flow characteristics inside the rectangular section air ejector are studied from the aspects of velocity distribution, turbulent kinetic energy distribution, Mach number distribution, and vorticity distribution.

### 3.1. Velocity Distribution of Rectangular Section Air Ejector

The rectangular section air ejector does not have rotational symmetry. Therefore, the analysis method of the rectangular section air ejector is different from that of the conventional circular section air ejector. The XOY plane and the XOZ plane of the rectangular section air ejector were selected to analyze the velocity distribution, as shown in Figure 5 and Figure 6. As is shown in Figure 5, the length of the central jet from the nozzle gradually increases with the increase of *W_c_*. When *W_c_* reaches 5 mm, the length of the central jet reaches a maximum, and the flow of the central jet is in a stable state. As *W_c_* continues to increase, flow separation occurs at the tail of the central jet. When *W_c_* increases to 7 mm, the central jet will expand further and produce a droplet jet at the inlet of the diffuser. This is caused by the sudden change of the structure at the inlet of the diffuser, which causes the velocity of the central jet suddenly increasing at the end of the constant-area section. Furthermore, due to the restrictions of the wall, the central jet will have a violent collision with the wall in the process of flow when *W_c_* is 1 mm. The friction between the central jet and the wall leads to the decrease of the central jet velocity, which is more obvious at the axis of the central jet. Figure 6 shows the velocity distribution of the rectangular section air ejector in the XOZ plane under different *W_c_*. Due to the limitation of the wall structure, the central jet cannot expand freely and the velocity gradient of the central jet also changes. When *W_c_* is 1 mm or 2 mm, the velocity of the central jet in the constant-area section is lower than that of other structures. With the increase of *W_c_*, the velocity distribution area of the central jet expands gradually.

### 3.2. Turbulent Kinetic Energy Distribution of Rectangular Section Air Ejector

Figure 7 is the turbulent kinetic energy (*TKE*) distribution of the rectangular section air ejector in the XOY plane under different *W_c_*. As demonstrated in Figure 7, with the increase of *W_c_*, the region of the *TKE* gradually enlarges. In the mixing chamber, the energy exchange between the primary fluid and the secondary fluid is mainly in the form of turbulent diffusion. As *W_c_* increases, more energy is exchanged. When *W_c_* increases to 5 mm, the *TKE* in the constant-area section will no longer increase. The energy transfer process of the two fluids is stable. At this time, the energy exchange between the two fluids reaches a stable stage.

Figure 8 shows the *TKE* distribution of the rectangular section air ejector in the XOZ plane with different *W_c_*. Figure 8 reveals that the *TKE* near the nozzle outlet fluctuates strongly when *W_c_* is 1 mm. When *W_c_* increases to 2 mm, the fluctuation region of the turbulent kinetic energy moves downstream, but the fluctuation of the *TKE* is still higher than that of other structures. Meanwhile, the mixing of the two fluids occurs closer to the inlet of the constant-area section. With the increase of *W_c_*, the mixing area of the two fluids gradually expands and moves to the outlet of the constant-area section. When *W_c_* increases to 5 mm, the fluctuation of the *TKE* occurs close to *X*_*c*1/2_, and the fluctuation only occurs near the wall. As the *W_c_* continues to increase, the primary fluid entrains the secondary fluid to the downstream of the constant-area section, where the mixing of the two fluids is gradually enhanced. At the same time, the fluctuation region of the *TKE* in the constant-area section will decrease until it disappears completely.

Figure 9 shows the *TKE* distribution of the rectangular section air ejector along the X axis under different *W_c_*. Figure 9 indicates that the *TKE* gradually weakens with the development of the fluids in the ejector. With the increase of *W_c_*, the region with the strongest turbulent kinetic energy gradually moves to *X_c_*_0_, and the shape of the fluctuation region of the *TKE* gradually tends to be consistent. It should be noted that the wall constraint on the fluid flow in the rectangular air ejector will be weakened with the increase of *W_c_*, but the dimension of the rectangular section air ejector in the XOY plane is fixed, which also plays a limiting role on the fluid flow in the rectangular section air ejector.

### 3.3. Mach Number Distribution of Rectangular Section Air Ejector

When the primary fluid leaves the nozzle, the central jet expands and accelerates into the mixing chamber. In the mixing chamber, the central jet forms a shock train alternating with expansion and compression. As far as we know, the structure of the shock wave, depending on the distribution of Mach number, has a direct effect on the performance of rectangular section air ejector. Therefore, the Mach number distribution of the rectangular section air ejectors under different *W_c_* was studied in this paper.

Figure 10 shows the Mach number distribution of the rectangular section air ejector in the XOY plane under different *W_c_*. As illustrated in Figure 10, the shock train length of the primary fluid is the shortest, due to the strongest limiting effect of the wall on the central jet when *W_c_* is 1 mm. As *W_c_* increases, the area of the central jet gradually increases, and the length of the shock train also gradually increases. When *W_c_* increases to 5 mm, the length of the central jet reaches a maximum, but the length of the shock train continues to increase. When *W_c_* is 9 mm, the length of the shock train reaches a maximum, and the *ER* of the air ejector also reaches its maximum. In addition, the shock train also forms a symmetric structure with the XOZ plane as the axis, which is called the double shock wave structure. This has not been observed in the conventional circular section air ejector. With the increase of *W_c_*, the double shock wave structure gradually moves closer to the XOZ plane. As *W_c_* increases to 7 mm, it begins to coincide at the front of the shock train. However, when *W_c_* increases to 10 mm, the dual shock structure moves away from the XOZ plane. For further analysis, the Mach number distribution of the rectangular section air ejector in the XOZ plane under different *W_c_* was obtained, as shown in Figure 11. As can be seen from Figure 11, there is no obvious shock train in the ejector when *W_c_* is 1 mm. With the increase of *W_c_*, the shock train gradually appears in the ejector, and the length of the shock train gradually becomes longer. This is caused by the gradual approach of the double shock wave structure to the XOZ plane. When *W_c_* increases to 10 mm, the length of the shock train in the ejector becomes shorter again, indicating that the dual shock wave structure is further away from the XOZ plane.

### 3.4. Vorticity Distribution of Rectangular Section Air Ejector

Vorticity is an important parameter to characterize the degree of turbulence. Under the restriction of *W_c_*, the high-speed central jet impinges on the wall in the rectangular section air ejector. The vortex in the ejector gradually twists and stretches, expands to all directions in space, and forms a vorticity field. The vorticity field includes streamwise vortices and spanwise vortices. The streamwise vortices play an important role in the entrainment process between the primary fluid and the secondary fluid. The spanwise vortices are generated by the velocity gradient between the primary fluid and the secondary fluid, which is mainly distributed near the nozzle outlet. The streamwise vortices (Ω*_s_*) and the spanwise vortices (Ω*_n_*) are defined as follows [35]:(10)$$\Omega_{s}=\frac{D_{0}}{U_{0}}(\frac{\partial w}{\partial y}-\frac{\partial v}{\partial z})$$
(11)$$\Omega_{n}=\frac{D_{0}}{U_{0}}\sqrt{{⟮\frac{\partial u}{\partial z}-\frac{\partial w}{\partial x}⟯}^{2}+{⟮\frac{\partial v}{\partial x}-\frac{\partial u}{\partial y}⟯}^{2}}\;$$
where *D*_0_ is the diameter of the nozzle outlet, and *U*_0_ is the mean velocity of the primary fluid the inlet of the nozzle.

The rotational motion of the streamwise vortices causes a relative flow between the primary fluid and the secondary fluid, which increases the mixing effect of the two fluids. Therefore, the influence of *W_c_* on the streamwise vortices in the rectangular section air ejector was studied in this paper. Figure 12 shows the streamwise vortices distribution of the rectangular section air ejector along the X-axis under different *W_c_*. In the figure, the red region is the positive vorticity, rotating in a counterclockwise direction. Meanwhile, the blue region is the negative vorticity, which rotates in a clockwise direction. In Figure 12, it can be seen how the motion of the streamwise vortices cause the secondary fluid to flow inward at the initial stage of the mixing process. With the development of fluid flow, the streamwise vortices at the center diffuse toward the wall, and the intensity of the streamwise vortices gradually weakens. Due to the sudden change of the structure of the rectangular section air ejector at the inlet of the diffuser, the intensity of the streamwise vortices increases sharply at the end of the constant-area section. When *W_c_* is 1 mm, the streamwise vortices at *X_c_*_0_ almost fill the entire section, and the wall attachment phenomenon of the streamwise vortices is the most obvious due to the limitation of rectangular flow channel. This wall attachment phenomenon is not conducive to the mixing of the two fluids, resulting in the lowest *ER* of the rectangular section air ejector under this structure. With the increase of *W_c_*, the streamwise vortices at *X_c_*_0_ begin to weaken. Simultaneously, the streamwise vortices develop gradually along the XOY plane, promoting the mixing of the two fluids.

Driven by viscous shear forces, the spanwise vortices are generated by the velocity gradient of the primary fluid and the secondary fluid along the edge of the geometric structure. The spanwise vortices are the vortex component perpendicular to the streamwise vortices. During the development of fluid flow, the spanwise vortices interact with the streamwise vortices. At the initial stage of the mixing process, the velocity gradient between the primary fluid and the secondary fluid is the largest, and the vortex structure near the nozzle is mainly the spanwise vortices. With the development of fluid flow, the spanwise vortices near the nozzle outlet begins to expand toward the wall and gradually weakens. Figure 13 is the spanwise vortices distribution of the rectangular section air ejector along the X-axis under different *W_c_*. In Figure 13, it can be seen how the distribution of the spanwise vortices is most obviously constrained by the wall when *W_c_* is 1 mm. With the increase of *W_c_*, the spanwise vortices structure near the nozzle outlet began to expand towards the wall and gradually weakened. When *W_c_* increases to 5 mm, the constraint of the wall on the spanwise vortices is weakened. As *W_c_* continues to increase, the variation of the spanwise vortices in the constant-area section is no longer obvious.

## 4. Conclusions

In order to investigate the influence of *W_c_* on the ER and the fluid flow characteristics of the rectangular section air ejector when the primary fluid pressure was 400 kPa and the secondary fluid pressure was 100 kPa, the velocity distribution, the turbulent kinetic energy distribution, the Mach number distribution, and the vorticity distribution in the rectangular section air ejector were studied by adopting the three-dimensional numerical simulation method. The following conclusions were obtained:(1)With the increase of *W_c_*, the *ER* of the rectangular section air ejector first increases rapidly and then fluctuates slightly. When *W_c_* increases from 1 mm to 10 mm, the minimum *ER* is 0.34, the maximum *ER* is 0.65, and the increment of the *ER* is 91.2%.(2)With the increase of *W_c_*, the distribution of the *TKE* gradually expands. In the mixing chamber, the energy exchange between the primary fluid and the secondary fluid is mainly in the form of turbulent diffusion. When *W_c_* increases to 5 mm, the *TKE* in the constant-area section no longer increases. Currently, the energy exchange between the two fluids reaches a stable stage. As *W_c_* continues to increase, the primary fluid entrains the secondary fluid to the downstream of the constant-area section, and the mixing of the two fluids gradually increases in the downstream. In addition to *W_c_* limiting the fluid flow in the rectangular section air ejector, the dimension of the rectangular section air ejector in the XOY plane also has a limiting effect on the fluid flow in the rectangular section air ejector.(3)With the increase of *W_c_*, the region of the central jet gradually increases, as does the length of the shock train. When *W_c_* increases to 5 mm, the length of the central jet reaches a maximum, but the length of the shock train continues to increase. When *W_c_* is 9 mm, the length of the shock train reaches a maximum, and the *ER* of the rectangular section air ejector also reaches a maximum.(4)In the rectangular section air ejector, the streamwise vortices play a primary role in the mixing process. Due to the limitation of *W_c_*, the mixing effect caused by the streamwise vortex is weakened, and the loss of the two fluids increases in the energy exchange process. Increasing *W_c_* will increase the distribution of the streamwise vortices in the constant-area section, and simultaneously, the distribution of the spanwise vortices will gradually decrease.

## Figures and Tables

**Figure 1 entropy-25-00179-f001:**
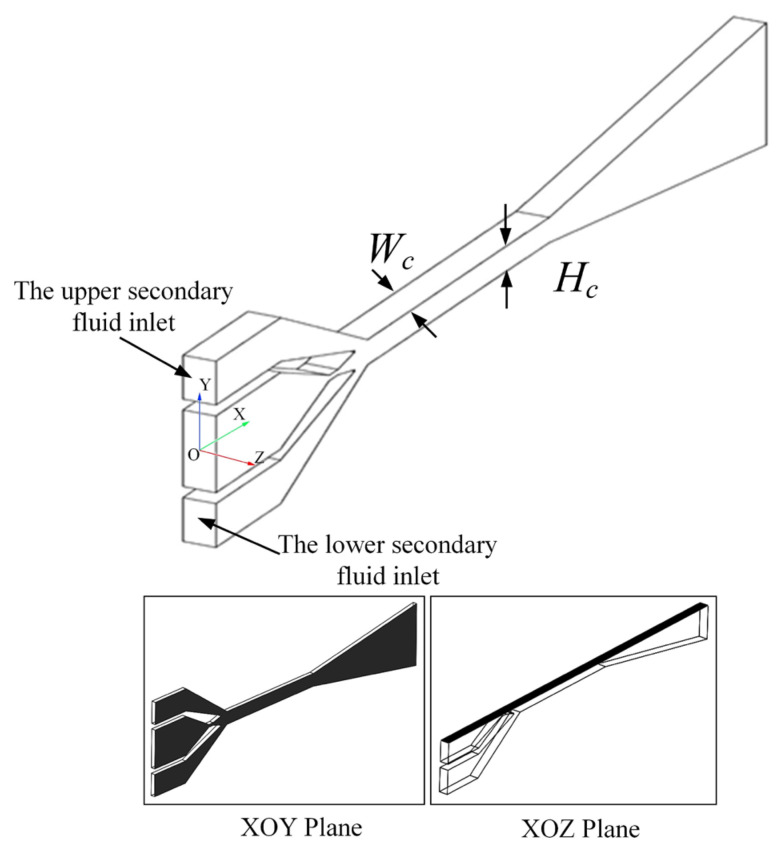
Three-dimensional structure diagram of the rectangular section air ejector.

**Figure 2 entropy-25-00179-f002:**
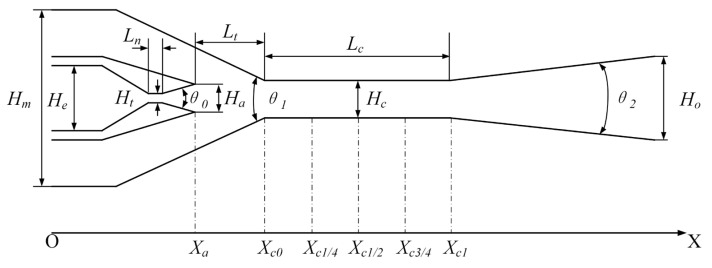
Structure diagram of the rectangular section air ejector on the XOY plane.

**Figure 3 entropy-25-00179-f003:**
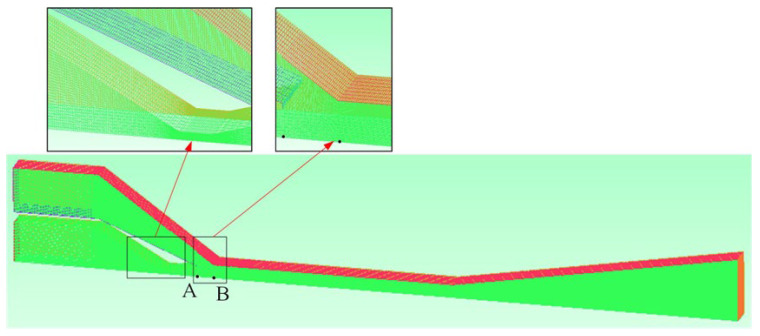
Three-dimensional grid diagram of the rectangular section air ejector.

**Figure 4 entropy-25-00179-f004:**
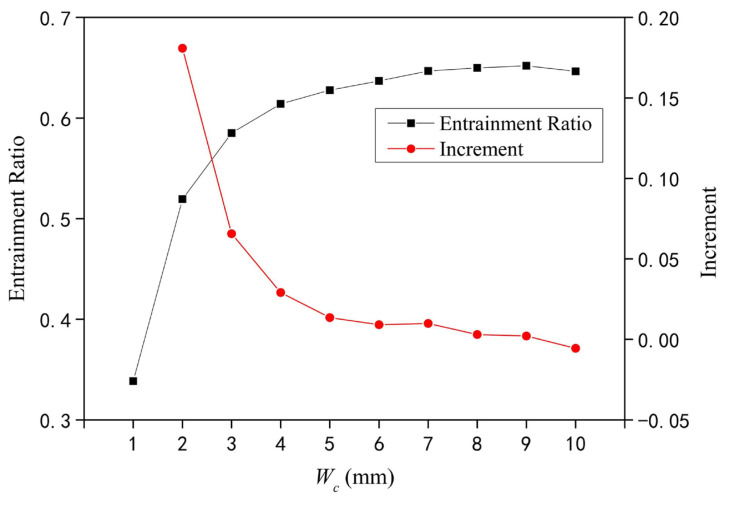
Variation of the *ER* of the rectangular section air ejector under different *W_c_*.

**Figure 5 entropy-25-00179-f005:**
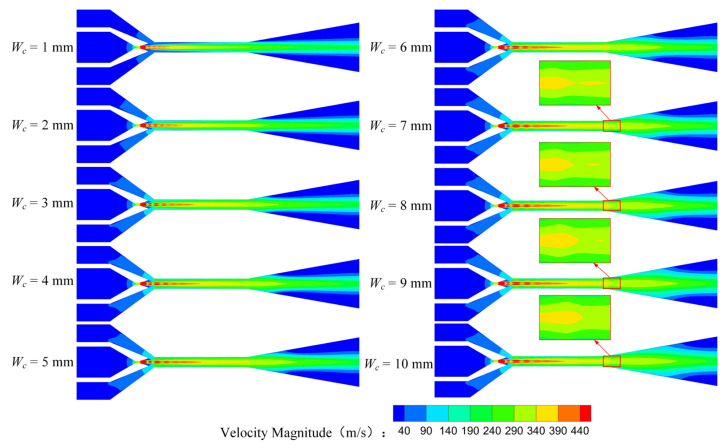
Variation of velocity distribution of the rectangular section air ejector in the XOY plane under different *W_c_*.

**Figure 6 entropy-25-00179-f006:**
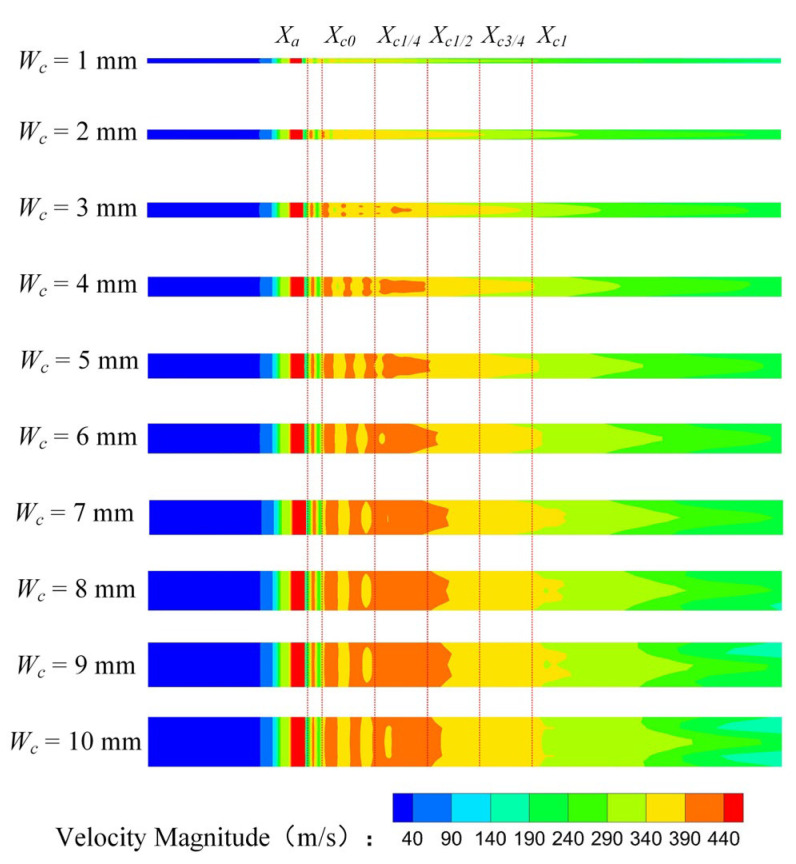
Variation of velocity distribution of the rectangular section air ejector in the XOZ plane under different *W_c_*.

**Figure 7 entropy-25-00179-f007:**
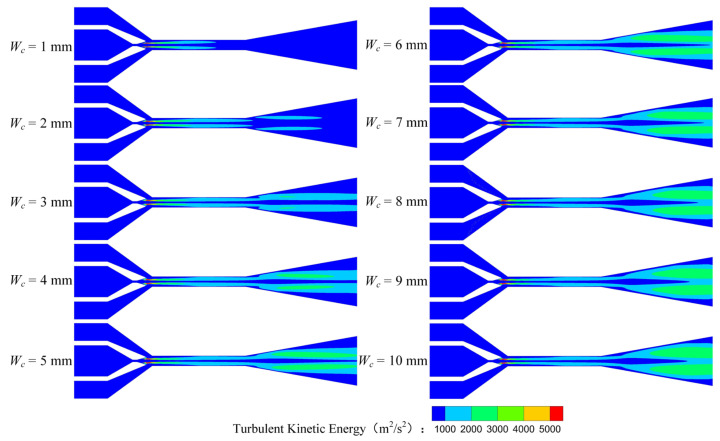
Variation of turbulent kinetic energy distribution of the rectangular section air ejector in the XOY plane under different *W_c_*.

**Figure 8 entropy-25-00179-f008:**
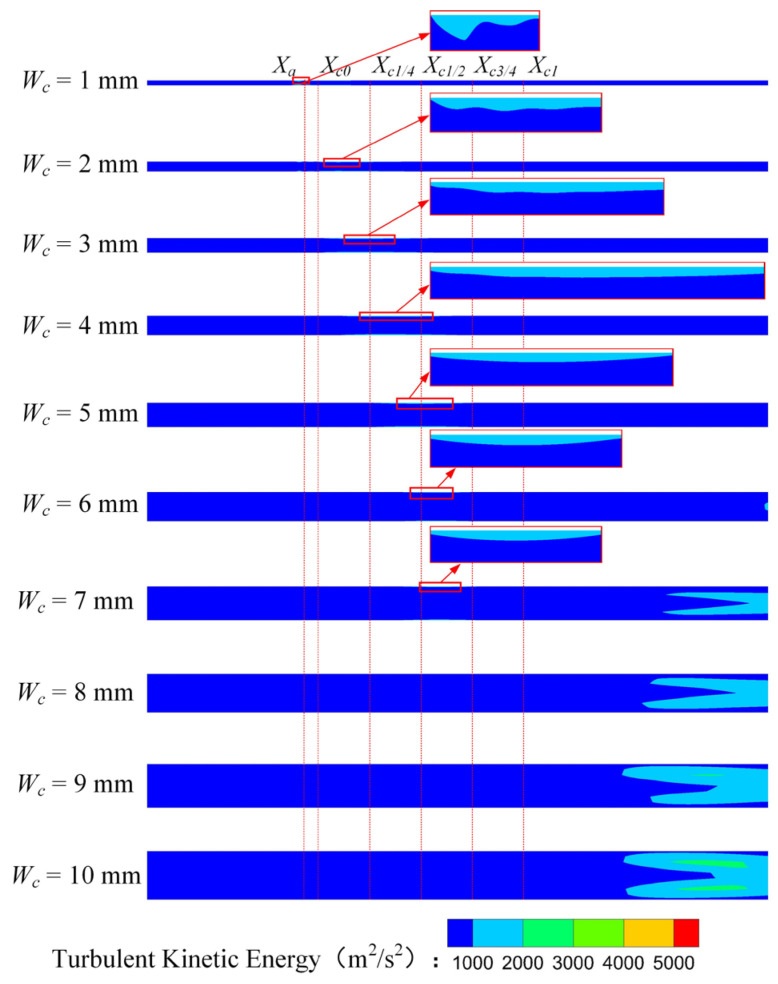
Variation of turbulent kinetic energy distribution of the rectangular section air ejector in the XOZ plane under different *W_c_*.

**Figure 9 entropy-25-00179-f009:**
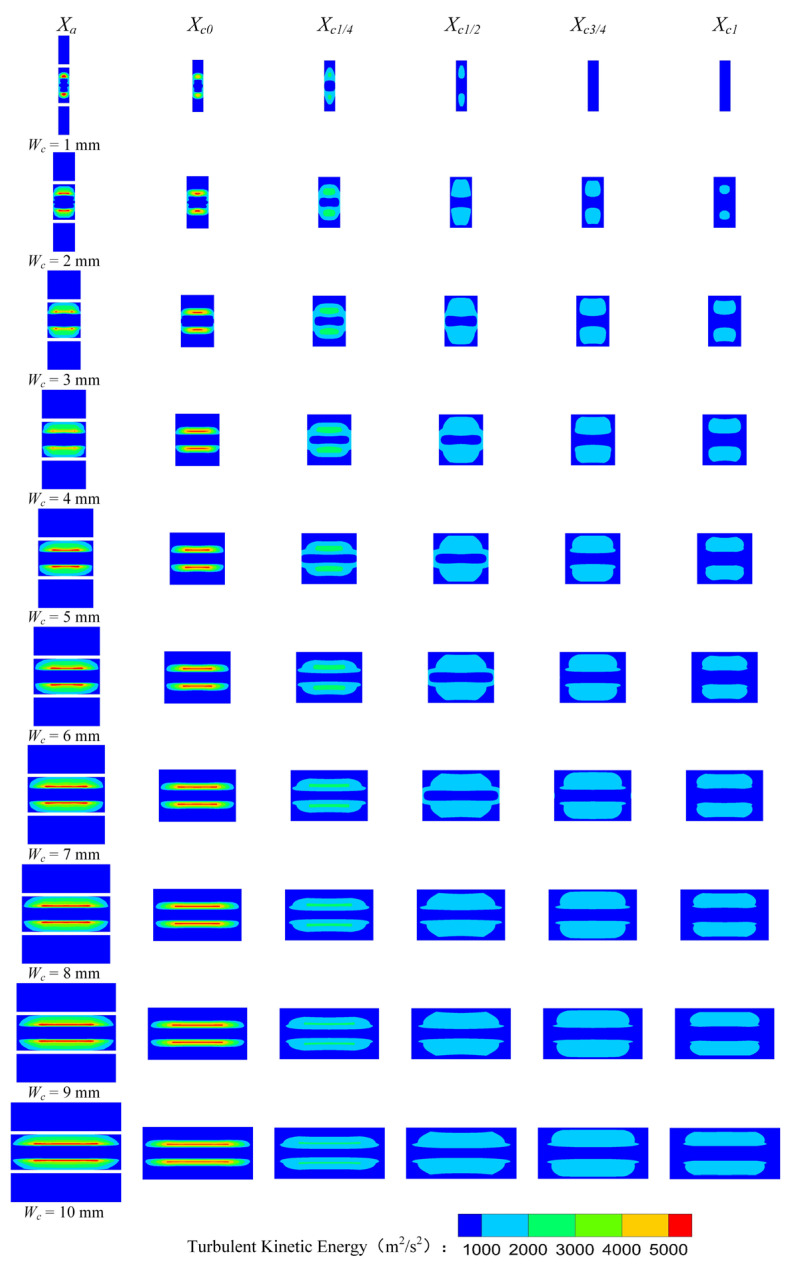
Variation of the turbulent kinetic energy distribution of the rectangular section air ejector along the X axis under different *W_c_*.

**Figure 10 entropy-25-00179-f010:**
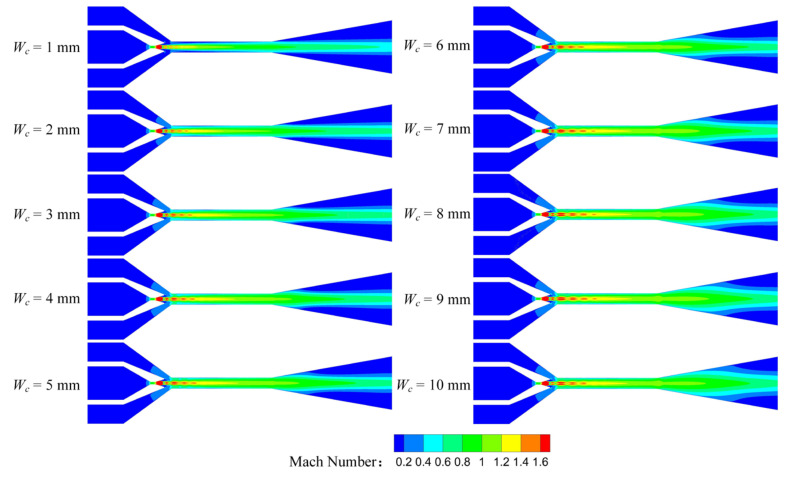
Variation of the Mach number distribution of the rectangular section air ejector in the XOY plane under different *W_c_*.

**Figure 11 entropy-25-00179-f011:**
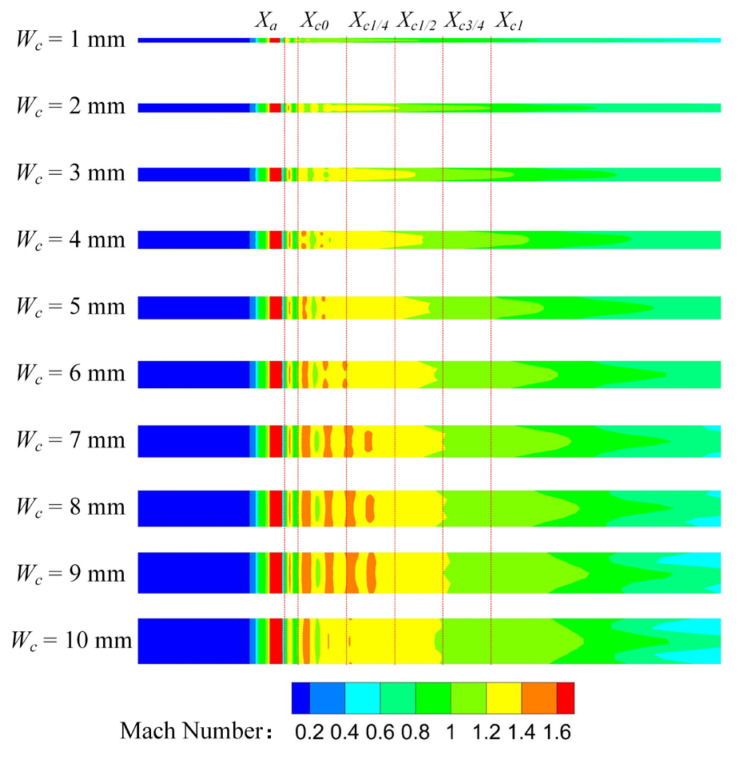
Variation of the Mach number distribution of the rectangular section air ejector in the XOZ plane under different *W_c_*.

**Figure 12 entropy-25-00179-f012:**
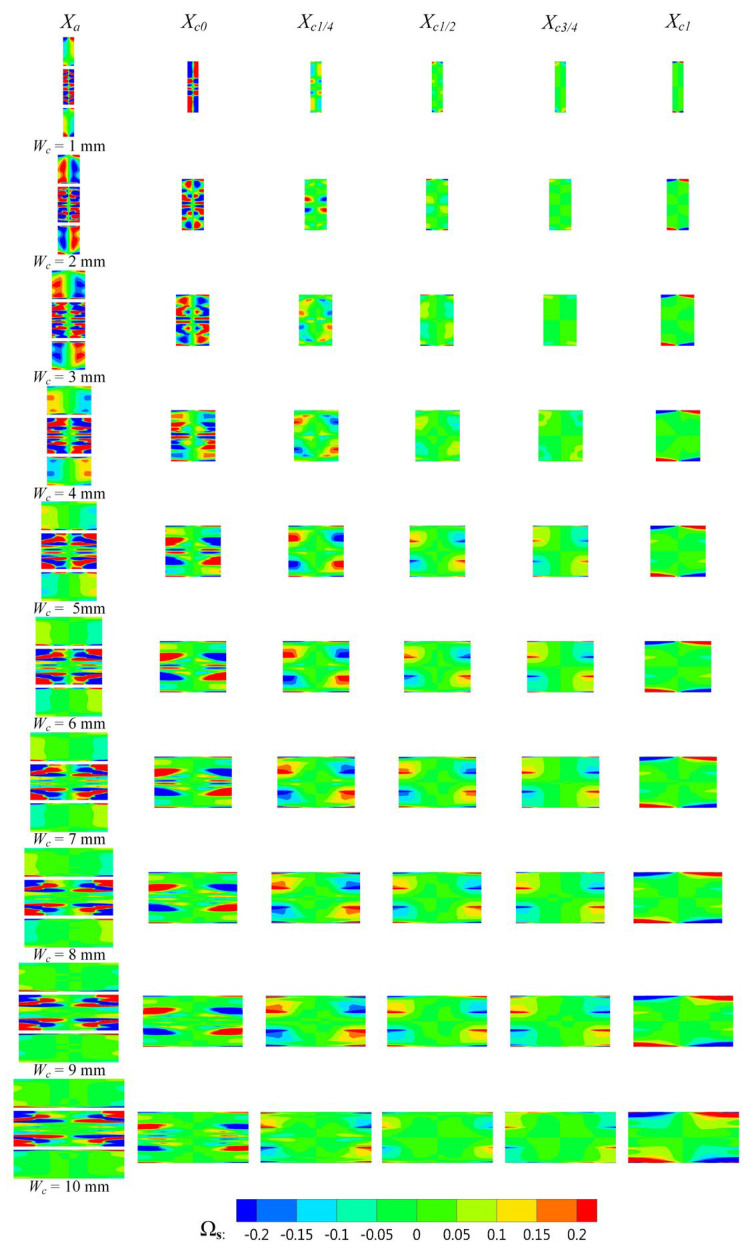
Variation of the streamside vortex distribution of the rectangular section air ejector along the X axis under different *W_c_*.

**Figure 13 entropy-25-00179-f013:**
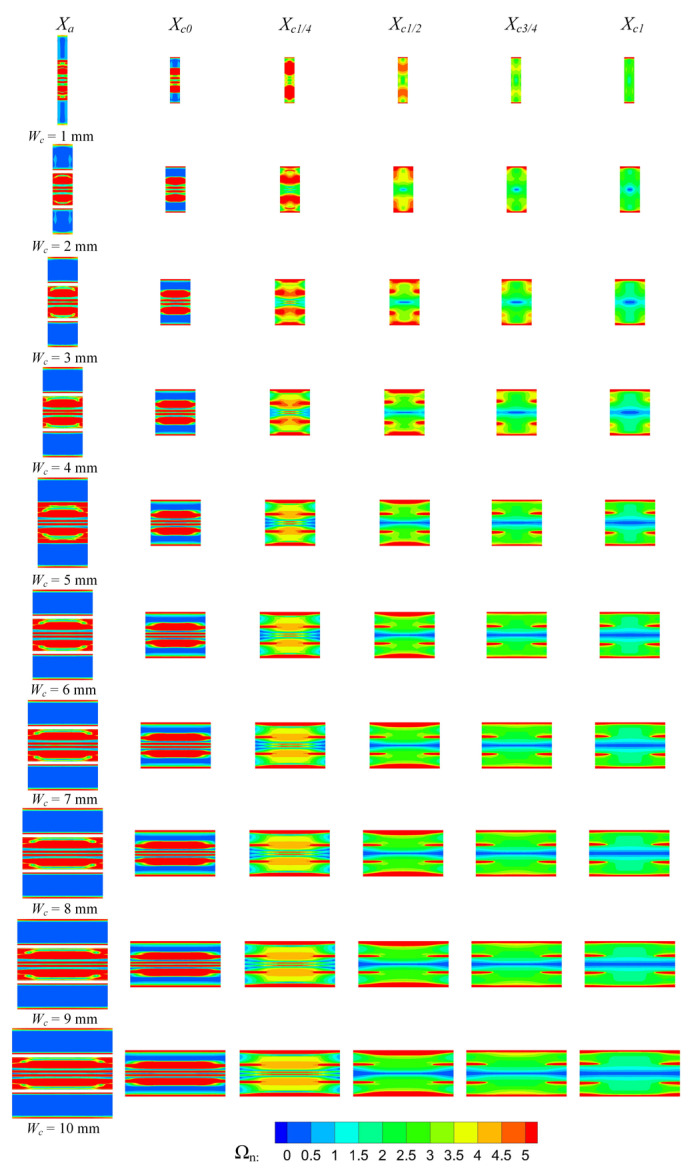
Variation of the spanwise vortices distribution of the rectangular section air ejector along the X axis under different *W_c_*.

**Table 1 entropy-25-00179-t001:** Structure parameters of the rectangular section air ejector.

Parameter Description	Symbol	Value	Units
Height of nozzle outlet	*H_a_*	3.2	mm
Height of constant-area section	*H_c_*	4.6	mm
Height of nozzle inlet	*H_e_*	14	mm
Height of mixing chamber inlet	*H_m_*	34	mm
Height of diffuser outlet	*H_o_*	22.3	mm
Height of nozzle throat	*H_t_*	1	mm
Length of constant-area section	*L_c_*	42	mm
Length of nozzle throat	*L_n_*	2	mm
Length of nozzle outlet to constant-area section inlet	*L_t_*	3	mm
Section width	*W_c_*	1, 2, 3, 4, 5, 6, 7, 8, 9, 10	mm
Divergent angle of nozzle	*θ* _0_	32	°
Convergent angle of mixing chamber	*θ* _1_	72	°
Divergent angle of diffusion chamber	*θ* _2_	20	°

**Table 2 entropy-25-00179-t002:** Grid independence test.

	Grid Numbers	Velocity (m/s)	Deviation (%)	Pressure (kPa)	Deviation (%)
Point A	148,864	582.97		22.17	
	213,634	582.78	−0.0326	22.23	0.27
	279,910	582.55	−0.0395	22.31	0.36
	503,690	582.54	−0.0017	22.31	0
	842,258	582.54	0	22.31	0
Point B	148,864	422.57		100.09	
	213,634	422.74	0.0402	99.76	−0.33
	279,910	422.92	0.0426	99.56	−0.20
	503,690	423.01	0.0213	99.10	−0.46
	842,258	423.01	0	99.10	0

## Data Availability

The research data supporting this publication are provided within this paper.

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
