# Peer review of "Numerical Investigation on the Effect of Section Width on the Performance of Air Ejector with Rectangular Section"

_entropy, 2023, doi:10.3390/e25010179_

Round 1

Reviewer 1 Report

This paper performs a numerical investigation on the effect of section width on the performance of an air ejector. The results are meaningful and generally well-presented. Nevertheless, there are some issues to address before I can recommend its publication:

Introduction: l. 84-88. It is not clear if the novelty of the paper lies in the variables studied or in other aspects of the paper.

Line: 91. Could you elaborate further the one- dimensional constant pressure mixing theory and provide a bit more information about the design of the air ejector?

2.2. CFD modelling:

-What was the CFD software employed to solve the equations?

- What was the discretization scheme for pressure?

- Grid independence study: what was the max and average y+ value in the meshes? This is important to include, as the validity of the turbulence model

Results

- It would be useful to draw the XYZ axes in the figures.

-“Furthermore, due to the restrictions of the wall, the central jet will have a violent collision

with the wall in the process of flow when Wc is 1 mm. The friction between the central jet

and the wall promotes the flow separation of the jet, which is more obvious in the low-

speed zone of the central”.  Separation would be better shown with axial velocity rather than velocity magnitude contours.

Reviewer 2 Report

In this study, numerical simulation has been performed on a non-circular (rectangular) ejector, and the effect of section width on the performance has been computed along the Mach number, velocity distribution, turbulent kinetic energy, and vorticity visualization along the ejector. I appreciate the authors for studying and exploring rectangular ejectors. Minor comments and suggestions are as follows to improve the manuscript further.

1.       In the introduction author should include different other design approaches of ejectors viz, constant area mixing (CAM), Constant Rate of Momentum Change (CRMC), and Constant Rate of Kinetic Energy Change (CRKEC) and justify the selection of the CPM approach for in the current study. For reference, the author can refer to these articles https://doi.org/10.1016/j.applthermaleng.2013.06.045

https://doi.org/10.1016/j.energy.2018.08.184

2.       At what basis the mentioned structural parameters (Table 1) were computed?  Also, mention the assumption made.  

3.       As per Figure 2, the nozzle throat has a constant length. Is it correct? If yes, Please mention the length of the nozzle throat in Table 1.

4.       Golable performance parameter entrainment ratio should be considered for the grid independence test.

5.       In Table 2, Instead of writing Error, you should calculate deviation as the exact value is unknown.

6.       Fig 4 shows the increment decreases from 2 mm onwards. While the author has written from 5 mm, kindly verify line 188.

7.       For better visualization of vorticities and their direction, post-processing of results is required. From the current figure, it is difficult to say the direction of vorticities in the flow domain.

8.       Authors should also incorporate quantitative results in discussing results.

9. Check references 5 and 9. 

Round 2

Reviewer 2 Report

Author has revised and incorporated suggestions made by me.